

# Speciated and total emission factors of particulate organics from burning western U.S. wildland fuels and their dependence on combustion efficiency

Coty N. Jen[1,6*], Lindsay E. Hatch[2], Vanessa Selimovic[3], Robert J. Yokelson[3], Robert Weber[1], Arantza
E. Fernandez[4], Nathan M. Kreisberg[4], Kelley C. Barsanti[2], Allen H. Goldstein[1,5]

[1] Department of Environmental Science, Policy, and Management, University of California, Berkeley, Berkeley, CA, 94720, USA

[2] Department of Chemical and Environmental Engineering and College of Engineering – Center for Environmental Research and Technology, University of California, Riverside, Riverside, CA, 92507, USA

[3] Department of Chemistry, University of Montana, Missoula, 59812, USA

[4]Aerosol Dynamics Inc., Berkeley, CA 94710, USA

[5]Department of Civil and Environmental Engineering, University of California, Berkeley, Berkeley, CA 94720, USA

[6]now at Department of Chemical Engineering, Carnegie Mellon University, Pittsburgh, PA 15213, USA

*correspond author: cotyj@andrew.cmu.edu

**Keywords:**

Biomass burning, FIREX, emission factors, I/SVOCs, MCE

**Abstract:** Western U.S. wildlands experience frequent and large-scale wildfires which are predicted to increase in the future. As a result, wildfire smoke emissions are expected to play an increasing role in atmospheric chemistry while negatively impacting regional air quality and human health. Understanding the impacts of smoke on the environment is informed by

identifying and quantifying the chemical compounds that are emitted during wildfires and by providing empirical relationships that describe how the amount and composition of the emissions change based upon different fire conditions and fuels. This study examined particulate organic compounds emitted from burning common western U.S. wildland fuels at the U.S. Forest Service Fire Science Laboratory. Thousands of intermediate and semi-volatile organic compounds (I/SVOCs) were separated and quantified into fire-integrated emission factors (EFs) using thermal desorption, two-dimensional gas

chromatograph with online derivatization coupled to an electron ionization/vacuum ultra-violet high-resolution time of flight mass spectrometer (TD-GC×GC-EI/VUV-HRToFMS). Mass spectra, EFs as a function of modified combustion efficiency (MCE), fuel source, and other defining characteristics for the separated compounds are provided in the accompanying mass spectral library. Results show that EFs for total organic carbon (OC), chemical families of I/SVOCs, and most individual I/SVOCs span 2-5 orders of magnitude, with higher EFs at smoldering conditions (low MCE) than flaming. Logarithmic fits

applied to the observations showed that log(EF) for particulate organic compounds were inversely proportional to MCE.



These measurements and relationships provide useful estimates of EFs for OC, elemental carbon (EC), organic chemical families, and individual I/SVOCs as a function of fire conditions.

## 1 Introduction

Wildfires in the western U.S. have become larger and more frequent, and this trend is expected to continue in the coming

decades (Dennison et al., 2014; Miller et al., 2009). This is due to historical wildfire suppression leading to high fuel loading and climate changes that include longer springs and summers, earlier snow melts, and prolonged droughts (Dennison et al., 2014; Jolly et al., 2015; Spracklen et al., 2009; Westerling et al., 2006). Smoke emissions from wildfires primarily contain carbon dioxide ($CO_2$), carbon monoxide (CO), and thousands of organic compounds in the gas and particle phases. These organic compounds can significantly influence atmospheric chemistry, cloud formation, regional visibility, and human

health. Thus, increased occurrences and magnitudes of wildfires will likely lead to greater smoke impacts on regional and global environments.

The extent to which smoke will adversely impact human health and the environment depends, in part, on the chemical composition and amount of emissions produced. In general, biomass burning, which includes wildfires, is the main global source of fine carbonaceous aerosol particles (~75%) in the atmosphere (Andreae and Merlet, 2001; Bond and Bergstrom,

2006; IPCC, 2014; Park et al., 2007). Individual and categorized organic emissions from wildfires have previously been identified and quantified (Akagi et al., 2011; Andreae and Merlet, 2001; Hatch et al., 2015; Kim et al., 2013; Koss et al., 2018; Liu et al., 2017; Mazzoleni et al., 2007; Naeher et al., 2007; Oros et al., 2006; Oros and Simoneit, 2001a; Simoneit, 2002; Stockwell et al., 2015; Yokelson et al., 2013). The bulk of previous studies on speciated organic compound emissions focused on gas-phase volatile organic compounds (VOCs). Particle-phase results are typically reported as total organic

carbon (OC) or particulate matter (PM) with aerodynamic diameters less than 10 or 2.5 μm ($PM_{10}$ and $PM_{2.5}$). These types of measurements provide no chemical specificity of the particle phase and thus limits the ability to predict how smoke will age in the atmosphere and impact the environment.

Several studies have examined specific particle-phase organic compounds in smoke such as toxic retene and other polycyclic aromatic hydrocarbons (PAHs) (e.g., Jayarathne et al., 2018; Kim et al., 2013; Naeher et al., 2007; Sullivan et al., 2014) or

abundant tracer compounds, like levoglucosan and vanillic acid (Simoneit et al., 1999). Out of the likely thousands of unique compounds, roughly 400 known particle-phase organic compounds and their amounts produced per mass of dry fuel burned (a quantity known as the emission factor, EF) have been published and organized by wildland type (Oros et al., 2006; 2001a; 2001b). These compounds span many chemical families (i.e., functionalities), like sugars and methoxyphenols, and provide key insights into how different wildland burns lead to different organic particulate composition and EFs.

New advances in instrumentation, such as two-dimensional gas chromatography or electrospray ionization coupled to high resolution mass spectrometers (Isaacman et al., 2011; Laskin et al., 2009), now allow for unprecedented levels of molecular





speciation of atmospheric aerosol particles that can further identify and quantify the thousands of previously unreported biomass burning compounds. Nevertheless, current fire and atmospheric chemistry models that predict amount of smoke produced, its atmospheric transformation/transportation, and its physiochemical properties (e.g., French et al., 2011; Reinhardt et al., 1997; Wiedinmyer et al., 2011) do not model the thousands of organic compounds emitted from fires due

primarily to limited computational resources. To address this deficiency, Alvarado et al. (2015) used measured volatility distribution bins of organic compounds from fresh smoke and modeled their atmospheric aging by assuming shifts in volatility distribution via reactions with ozone and hydroxl radicals. Though this approach does better predict secondary organic aerosol particle formation, it still does not consider the wide variety of chemical compounds found in smoke, thus limiting its ability to predict physiochemical properties of aged smoke particles and their impacts on the environment.

Therefore, to better represent the chemical composition of smoke particles in models requires condensing the information from molecular-level speciation into useable relationships that correlate typical particle composition to a measurable burn variable.

The purpose of this study is to (1) identify, classify, and quantify organic compounds in smoke particles produced during laboratory burns and (2) provide scalable EFs of individual compounds and their chemical families from various fuels as a

function of fire conditions. A selection of fuels and fuel combinations commonly consumed in western U.S. wildland fires was burned at the U.S. Forest Service Fire Sciences Laboratory (FSL) in Missoula, MT during the NOAA Fire Influence on Regional and Global Environments Experiment (FIREX) campaign in 2016 (Selimovic et al., 2018). Regression models representing EFs as a function of fire conditions are provided for groupings of organic compounds that vary in chemical complexity, from generalized organic carbon and total particulate organic compounds to specific chemical families and

individual compounds. In addition, a mass spectral database, compatible with the National Institute of Standards and Technology (NIST) Mass Spectral Search program, containing the mass spectra, retention indices, identities/compound classifications for all the separated compounds observed from the various burns is included. This database will be a valuable resource for the community for identifying specific chemicals in air masses impacted by biomass burning plumes and understanding the dominant source materials burned, fire characteristics, and atmospheric transformations.

**2 Materials and Methods**

Thirty-four different fuels were combusted in 75 "stack" burns during the 2016 FIREX campaign at FSL (Selimovic et al., 2018). Most fuels were representative of common biomass components found in the western U.S wildlands. Non-western U.S. wildland fuels were also burned and are used to demonstrate the applicability of the reported regression models across a wider range of fuels. A detailed description of the FSL combustion room can be found elsewhere (Christian et al., 2004;

Stockwell et al., 2014) with pertinent details described here. The 12.5 m × 12.5 m × 22 m combustion room contained a fuel bed on the floor. Fuels were placed in the fuel bed and ignited by resistance-heated coils. Above the fuel bed was a 3.6 m inverted funnel connected to a 1.6 m diameter exhaust stack that vented through the roof of the combustion room. The room



was kept at positive pressure to provide a constant air flow that diluted and carried the smoke up the stack. A platform was located 17 m above the fuel bed and allowed instrument sampling access into the stack (see Figure S1 in the supporting information, SI). The samples studied here were collected from the platform and thus represent fresh emissions.

Smoke from the stack was pulled through a custom-built sampler known as DEFCON, Direct Emission Fire CONcentrator
(see diagram Figure S2 in SI). DEFCON's inlet was a 20.3 cm × 1.3 cm OD stainless steel tube that reached 15.2 cm into the stack. 10.3 LPM of smoke was pulled through the inlet, with two 150 ccm flows branching off from the main sample flow. These low-flow channels went to 2 parallel flows consisting of a Teflon filter followed by sorbent tube for gas-phase sample collection; analysis of those samples will be described in future publications. The remaining 10 LPM was passed through a 1.0 μm cutoff cyclone before being sampled onto a 10 cm quartz fiber filter (Pallflex Tissuquartz). Total residence time was
~2 s. One quartz fiber filter collected both particles and likely low volatility gases for the duration of each fire which lasted ~5-50 minutes. A few fires were terminated "early" when a small amount of fuel and smoldering combustion remained. Prior to collection, filters were baked at 550°C for 12 hours and packed in similarly baked aluminum foil inside Mylar bags. The flows were monitored to ensure constant flow rates. Flow paths within DEFCON were passivated with Inertium® (Advanced Materials Components Express, Lemont, PA) which has been shown to reduce losses of oxygenated organics (Williams et
al., 2006). After each burn, the inlet of DEFCON was replaced with a clean tube and the remainder of the system was purged with clean air. A background filter sample was collected each morning prior to the burns to estimate background contributions from sampling components and room air.

29 fire-integrated smoke filter samples, including one from each specific fuel (with some replicates), were selected and analyzed using a thermal desorption, two-dimensional gas chromatograph with online derivatization coupled to an electron
ionization/vacuum ultra-violet ionization high-resolution time of flight mass spectrometer (TD-GC×GC-EI/VUV-HRToFMS) (Isaacman et al., 2012; Worton et al., 2017). A list of analyzed fuels are summarized in Table 1. Punched samples of each filter (0.21-1.64 cm²) were thermally desorbed at 320°C under a helium flow using a thermal desorption system (TDS3 and TDSA2, Gerstel). Desorbed samples were then mixed with gaseous derivatization agent, MSFTA (N-methyl-N-trimethylsilyltrifluoroacetamide). MSFTA replaces the hydrogen in polar hydroxyl, amino, and thiol groups with
trimethylsilyl group, creating a less polar and thus elutable compound. Derivatized samples then were focused on a quartz wool glass liner at 30°C (cooled injection system, CIS4, Gerstel) before rapid heating to 320°C for injection into the gas chromatograph (GC, Agilent 7890). GC×GC separation was achieved with a 60 m × 0.25 mm × 0.25 μm semi-nonpolar capillary column (Rxi-5Sil MS, Restek) followed by medium-polarity second dimension column (1 m × 0.25 mm × 0.25 μm, Rtx-200MS, Restek). A dual-stage thermal modulator (Zoex), consisting of a guard column (1 m × 0.25 mm, Rxi, Restek),
was used to cryogenically focus the effluent from the first column prior to heated injection onto the second column (modulation period of 2.3 s). The main GC×GC oven ramped at 3.5°C/min from 40°C to 320°C and was held at the final temperature for 5 min and the secondary oven ramped at the same rate from 90°C to 330°C and held for 40 min. Separated compounds were then ionized either by traditional EI (70 eV) or VUV light (10.5 eV). HRToFMS (ToFWerk) was used to



detect the ions and was operated with a resolution of 4000 and transfer line and ionizer chamber temperatures at 270˚C. VUV light was provided by the Advanced Light Source, beamline 9.0.2, at Lawrence Berkeley National Laboratories. During the VUV experiments, the HRToFMS operated at a lower ionizer chamber temperature of 170˚C to further reduce molecular fragmentation (Isaacman et al., 2012).

Punches from the filter samples were also analyzed for organic and elemental carbon (OC and EC respectively) using a Sunset Model 5 Lab OCEC Aerosol Analyzer following the NIOSH870 protocol in the Air Quality Research Center at the University of California, Davis. Thermal pyrolysis (charring) was corrected using laser transmittance laser. OC and EC were also measured on the background filters.

**2.1 Emission Factor Calculations**

The mass loadings for all separated compounds measured by TD-GC×GC-EI/VUV-HRToFMS were determined using a set of calibration curves. Full details of the data inversion process and associated uncertainties are provided in the SI with important steps outlined here. The TD-GC×GC-EI/VUV-HRToFMS responses to a wide range of standard compounds commonly found in biomass burning samples were measured at varying mass loadings to create calibration curves. Measured peaks from the filters were calibrated using a standard compound that exhibited similar first and second dimension retention

times and compound classification. For example, a sampled compound classified as sugar was quantified using the nearest sugar standard compound in the chromatogram. Unknown compounds were matched to the nearest eluting standard compound. The mass loadings of all observed compounds were then background subtracted; however, the mass on the background filter for all compounds was negligible. The compound's emission factor ($EF_{compound}$) was then calculated by normalizing mass loadings by background-corrected sampled $CO_2$ mass. This ratio was then multiplied by the corresponding

$EF_{CO2}$, as given in Selimovic et al. (2018). EFs for OC and EC were calculated similarly, using background-corrected OC and EC mass loadings.

**3 Results and Discussion**

**3.1 Emission factors of organic and elemental carbon**

OC and EC EFs were first related to the fire-integrated modified combustion efficiency (MCE). MCE reflects the mix of

combustion processes in the fire and is defined as background-corrected values of $CO_2/(CO_2+CO)$ (Akagi et al., 2011; Ward and Radke, 1993). MCE values near 1 indicate almost pure flaming, while values near 0.8 are almost pure smoldering with 0.9 representing a roughly equal mix of these processes. Figure 1(a) shows the EFs of OC and EC ($EF_{OC}$ and $EF_{EC}$) as a function of MCE across a variety of fuel types (see table 1). Decreasing MCE (more smoldering) results in increased OC and decreased EC emissions across all studied fuel types. These observed trends are in general agreement with previous studies

(e.g., Christian et al., 2003; Hosseini et al., 2013). EFs for OC and EF generally follow a logarithmic relationship such that $\log(EF_{OC})$ is inversely proportional to MCE (slope of -13.9) and $\log(EF_{EC})$ is directly proportional (slope 8.3). Comparison





of the slopes suggests that decreasing MCE of a fire will produce an increasing amount of OC compared to EC. This is further confirmed by examining the ratio of OC to EC (OC/EC) with MCE. Figure 1(b) illustrates how OC/EC sharply increases with more smoldering fire conditions (slope of -24.6). This trend also follows a similar inversely proportional logarithmic relationship as $EF_{OC}$ vs. MCE but with even stronger correlation ($R^2$=0.85 compared to 0.66, respectively). Note,

values for Douglas Fir rotten log (burn 31), peat (burn 55), rice straw (burn 60), and Engelmann spruce duff (burn 26) fires are not shown due to measured EC at background levels. In addition, significant losses (~40%) of organic compounds were only observed for the Douglas Fir rotten log burn and was determined by comparing GC×GC chromatograms taken prior to OC/EC analysis (~2 years after collection) and ~1 month after collection at FSL. The combined results clearly show that flaming combustion produce slightly more particulate EC compared to OC whereas smoldering combustion emits 1-2 orders

of magnitude higher levels of OC compared to EC.

## 3.2  Identification and quantification of I/SVOCs:

Filter samples were analyzed for intermediate and semi-volatile organic compounds (I/SVOCs) using the TD-GC×GC EI-VUV-HRToFMS. Between 100-850 peaks (i.e., unique compounds) were separated in each fire-integrated chromatogram with fewer peaks observed for more flaming fires such as from shrub fuels (see Table 1). An example two-dimensional

chromatogram of a lodgepole pine burn (burn 63) is shown in Figure 2. Based on the GC×GC configuration, all compounds elute between dodecane ($C^*$~$10^6$ μg m$^{-3}$ and an $n$-alkane retention index, RI, of 1200) and hexatriacontane ($C^*$~$10^{-1}$ μg m$^{-3}$, RI=3600) and thus are classified as I/SVOCs with a small fraction as low-volatility organic compounds (Donahue et al., 2009). In total, approximately 3000 unique compounds were separated across the 29 analyzed burns (see  Table 1). From those compounds, 149 compounds were identified using a combination of matching authentic standards (STD), RI, EI mass

spectrum (via NIST mass spectral database, 2014 version), and VUV parent and fragment mass ions. A table of these identified compounds with their identifying methods (e.g., standard matching, previous literature, or NIST mass spectral database), RI, 5 most abundant mass ions, and fuel source(s) are given in Table S1.

To help reduce the chemical complexity from the 3000 observed compounds, each separated compound was sorted into a chemical family. This was achieved using a combination of parent ion mass (VUV), fragment ion mass spectra (VUV and

EI), RI, and second-dimension retention time to estimate the compound's functionality. More details on the classification process and examples within each category can be found in the SI. The chemical families were broadly named and include non-cyclic aliphatic/oxygenated, sugars, PAHs/methylated+oxygenated, resin acids/diterpenoids, sterols/ triterpenoids, organic nitrogen, oxygenated aromatic heterocycles, oxygenated cyclic alkanes, methoxyphenols, substituted phenols, and substituted benzoic acids. Almost 400 compounds, including the identified and most frequently observed compounds in the

analyzed burns, were grouped into these families. The remainder of the compounds, which were both uncategorizable and unidentifiable, were placed into the unknown category. Figure 2 illustrates the chemical families (indicated by color) of all the separated compounds emitted from an example lodgepole pine burn.



Despite many compounds remaining unknown, their defining traits such as mass spectra or retention index (i.e., volatility) can be compared to atmospheric samples to help the community better define the composition of biomass-burning derived, particle-phase organic compounds. As such, all ~3000 observed compounds have been compiled into a publicly available mass spectral database and first reported here as the University of California, Berkeley- Goldstein Library of Organic

Biogenic and Environmental Spectra (UCB-GLOBES) for FIREX (see SI). This spectral library is compatible with NIST MS Search and contains mass spectra, *n*-alkane RI, potential compound identification or chemical families, and fuel sources of all unique compounds detected from the 29 analyzed burns.

### 3.3  Average observed I/SVOC composition:

The masses of observed I/SVOCs from each chemical family were summed over each fire-integrated sample and normalized

to either the total observed I/SVOC mass or total classified I/SVOC mass. Figure 3a illustrates mass fractions of the unidentified and unclassified (unknown) compounds out of the total observed mass from the 29 analyzed burns. Unknowns represent ~35 to 90% of I/SVOCs mass emitted during the analyzed burns, with woody debris (rotten logs) exhibiting the highest mass fraction of unknowns (~90%). Since the compounds that make up the unknown mass fraction varied between burns, differences in the mass fractions between fuel types is not indicative of higher emissions of any particular compound.

However, notably the two woody debris burns showed similar unknown compounds (i.e., 99% of the unknown mass was of compounds found in both burns) but occurred under two different fire conditions (burn 13 at MCE=0.98 and burn 31 at MCE=0.78). In both cases, the unknown mass fractions were similar at 87-89%. This observation provides some indication that fuel type plays a larger role than MCE in determining the unknown organic mass fraction in smoke particles.

Given that the unknown compounds typically varied between burns, the mass of each classified chemical family was

normalized to the total observed classified mass (i.e., excluding the unknown mass) in order to better compare classified compounds between burns. These results are shown in Figure 3b. Conifers, coniferous litter, and wood exhibited the highest fraction of sugars (38%, 29%, and 44% respectively) compared to other fuels (between 6-30%). Furthermore, levoglucosan was the largest single contributor to the sugars for these burns and ranged from 10-40% of the total sugars. These observations are consistent with previous studies that have shown high levoglucosan emissions from cellulose-rich wood

samples (Mazzoleni et al., 2007; Simoneit et al., 1999). Coniferous fuels also emitted higher amounts of resin acids/diterpenoids (7%, 16%, 7%, and 3% for conifers, coniferous litter, coniferous duff, and woody debris respectively), as previously observed (Hays et al., 2002; Oros and Simoneit, 2001a; Schauer et al., 2001). Peat (from Indonesia) emitted the largest fraction of aliphatic compounds (52%) compared to other fuels, in agreement with previous observations (George et al., 2016; Iinuma et al., 2007; Jayarathne et al., 2018). Manzanita burns produced the highest amounts of substituted phenols

(34% of total classified mass compared to 1-4% for other fuels), mostly as hydroquinone (Hatch et al., In Prep. 2018; Jen et al., 2018). Organic nitrogen compounds, most of which were nitro-organics, also contributed significantly (up to 43%) to the total observed classified mass for all fuels. These compounds tend to absorb light (Laskin et al., 2015) and may contribute to



observed brown carbon light absorption from these burns (Selimovic et al., 2018). However, it should be noted that the instrument is not as sensitive to this class of compounds. Thus, the EF uncertainty is high (factor of 2) for compounds that are not positively identified with a standard but are categorized as organic nitrogen.

Figure 3b also provides some evidence that fuels within the same type generally show similar mass fractions of chemical families. For example, conifers, which consist of a mixture of coniferous ecosystem fuel component (e.g., canopy, duff, litter, and twigs), exhibit relatively similar mass fractions with sugars accounting for 30-50%, 4-28% non-cyclic aliphatic, 13-30% organic nitrogen, 2-20% resin acid/diterpenoids, and 1-4% PAH/methyl+oxy across the MCE range of 0.90-0.95. Coniferous duff (MCE=0.85-0.89) exhibited lower sugar fraction (11-31%) but higher non-cyclic aliphatics 15-38% than the conifers. Burning grasses (MCE=0.90-0.95) produced roughly equal amounts of sugars and organic nitrogen compounds (30%) and higher amounts of oxygenated cyclic compounds (3-11%), like lactones, than the coniferous fuels. (~1%). Shrubs (MCE=0.92-0.98) exhibited the largest ranges in chemical family mass fractions (e.g., 0-42% organic nitrogen compounds and 2-43% substituted phenols), suggesting that the fuels in this fuel type are less similar to each other when burned than pure coniferous fuels. Overall, the I/SVOC mass fractions tend to be more similar for fuels within a fuel type with the most variation for fuel mixtures and shrubs.

## 3.4 EFs as a function of fire conditions (MCE):

Unlike the dependence of chemical family mass fractions on fuel type, EFs for each chemical family showed a correlation with MCE across all fuels examined and to a much lesser extent on fuel type. Figure 4 presents EFs for total observed organic compounds and 5 of the chemical families (unknowns, sugars, PAHs/methyl/oxy, methoxyphenols, and sterols/triterpenoids, with others given in Figure S4) as a function of MCE. Fuels not found in the western U.S. are also included in these figures to demonstrate that their EFs generally follow the trend with MCE. The notable exception is peat, a semi-fossilized fuel (Stockwell et al., 2016), whose EFs for all chemical families are roughly an order of magnitude lower than other fuels at similar MCE values, except non-cyclic aliphatic/oxygenated EF which is approximately equal to burns at similar MCE (see Figure S4). In general, EFs measured by the TD-GC×GC-EI/VUV-HRToFMS agree with previous literature (Hays et al., 2002; McDonald et al., 2000; Oros et al., 2006; Oros and Simoneit, 2001a, 2001b). For example, Oros and Simoneit (2001a) provided the sum of carboxylic acids and alkanes/enes/ols for conifer burns at ~1 g/kg, which is within our reported range 0.4-2.3 g/kg (MCE=0.90-0.95) for non-cyclic aliphatic/oxygenated emitted from burning conifers. Other chemical family EFs presented here for conifers, including PAHs (0.4 g/kg), diterpenoids (1-3 g/kg), and methoxyphenols (1 g/kg), are also in good agreement with those published in Oros and Simoneit (2001a). Hays et al. (2002) reported EF for unknown compounds from ponderosa pine ~ 20 g/kg, higher than 11 g/kg (MCE=0.94) for the ponderosa pine burn studied here. This may be due to more compounds being classified here than in Hays et al. (2002) or differences in MCE between the studies. In contrast to previous work, EFs for chemical families reported here were measured over a wider range of fire conditions and fuel types and show a clear relationship with MCE.


Chemical family EFs span ~3 orders of magnitude and therefore logarithmic fits (given as a dashed line in the semi-log graphs of Figure 4) were applied to all the measurements excluding peat. Slopes range between -9.425 for sterols/triterpenoids to -14.637 for PAHs/methyl+oxy (fitted slopes, intercepts, and their errors for all chemical families are provided in Table S6). Three decimal places are provided for both the slope and intercept in order to reproduce the

regression line. The $R^2$ values for the sugars (Figure 4) and resin acids/diterpenoids (Figure S4) are noticeably lower at 0.32 and 0.31, respectively, than the other chemical families ($R^2$~0.4). This is primarily due to the high mass fraction of sugars and resin acids/diterpenoids found in conifers, as stated above, and suggests that high emissions of both types of compounds are indicative of burning conifers. Also, coniferous litter emitted high amounts of resin acids/diterpenoids. Removing conifers from the semi-logarithmic model for sugars yields a $\log_{10}(EF_{sugars})$= -10.299(MCE)+9.361 with $R^2$=0.66 and

removing conifers and coniferous litter for resin acids/diterpenoids results in $\log_{10}(EF_{resin})$= -19.598(MCE)+16.360 with $R^2$=0.77. These $R^2$ values are then more similar to those of other chemical families. In general, these results indicate that MCE can be used to estimate EFs for various chemical across of broad range of fuels, including those not found in the western U.S. wildlands except peat, with minimal dependence on fuel type.

The accuracy of the regression models can be evaluated by comparing predicted EFs to those measured in this study and

others. As evident in Figure 4a, the predicted total I/SVOC EFs are on average higher than the measured EFs by a factor of 2 overall, with shrubs by a factor 4, grasses factor of 2, conifers factor of 0.4, coniferous duff a factor of 2, and woody debris a factor of 0.6. Out of all the chemical families, the model over-predicts the most for the PAH EFs by an average factor of 3. The model is also compared to previously reported EFs from wildfires. Specifically, Liu et al. (2017) reported MCE values from three different California wildfires and total organic aerosol (OA) particle EFs, which are the most equivalent to total

I/SVOC EFs measured here (though at MCE values of <0.8, we observed higher IVOCs mass loadings in our chromatograms which would likely not be included in the OA EFs at lower particle mass loadings (May et al., 2013)). Liu et al. measured MCE values of 0.935, 0.877, and 0.923 with OA EFs of 23.3, 30.9, and 18.8 g/kg respectively. The model given in Figure 4a predicts total I/SVOC EFs of 8, 35, and 10 g/kg for those MCE values. The predicted EFs are within a factor of ~2-3, consistent with measured total I/SVOC EFs reported here. No other previous experiments report chemical

family EFs from wildfires with corresponding MCE values thus the accuracy of applying the chemical family regressions cannot be evaluated at this time. Without this information, uncertainty in using the reported regression models in predicting EFs of various chemical families is estimated to be a factor of 3. However, this uncertainty in EFs is minor when compared to uncertainties in estimating the amount of fuel burned in large-scale carbon emission fire models (French et al., 2011; Urbanski et al., 2011), which is primarily due to high spatial and temporal variations in fuel loadings and lack of

observational data. Thus, these regressions can be used to approximate EFs of various chemical families for a wide range of fuels and fuel mixtures from measured MCE values.

Figure 5 shows the fire-integrated EFs of four specific compounds, levoglucosan (sugar), fluoranthene (PAH), acetovanillone (methoxyphenol), and coniferyl aldehyde (methoxyphenol), as a function of MCE. Acetovanillone and



coniferyl aldehyde, both methoxyphenols, have been reported previously as tracers for lignin pyrolysis and levoglucosan (and more broadly sugars) from cellulose (Hawthorne et al., 1989; Oros and Simoneit, 2001a; Schauer et al., 2001; Simoneit, 2002). In addition, fluoranthene and other PAHs are known carcinogenic compounds (Boffetta et al., 1997; Kim et al., 2013). EFs for levoglucosan, the most widely reported particulate tracer compound for biomass burning (Mazzoleni et al., 2007;

Simoneit et al., 1999; Sullivan et al., 2014), range between ~0.004-1 g/kg from this study.  Hosseini et al. (2013) reported $EF_{levo}$ for chaparral ecosystems at 0.02-0.1 g/kg, similar to $EF_{levo}$ for shrubs measured (0.004-0.1) in this study. Schuaer et al. (2001) provided average $EF_{levo}$ for pine trees at 1.4 g/kg, roughly a factor of 2 higher than the average 0.6 g/kg $EF_{levo}$ for conifers of this study. Oros and Simoneit (2001a) examined levoglucosan emissions from various types of pine trees with an average $EF_{levo}$ of 0.02 g/kg, a factor of 30 lower than reported here. Many reasons could explain this difference, such as

different smoke sampling/filter extraction procedures and different MCE conditions during sampling. Regardless, the levoglucosan EFs reported in this study generally fall within the ranges measured by previous groups.

EFs for the compounds shown in Figure 5 span 2-5 orders of magnitude across fire conditions and fuels, including fuels found outside of the western U.S. Similar to the chemical family EFs, peat displays significantly lower EFs (factor of ~10) than the other fuels. Consequently, applied logarithmic fits, given as dashed lines in Figure 5, exclude peat. Slopes of these

fits range from -6.455 to -19.443 for the four displayed compounds. Figure 5a also shows that the $R^2$ value for levoglucosan is the lowest (0.34) compared to the other compounds (0.40-0.63). This poorer correlation with MCE is similar to that seen for the sugar EFs in Figure 4c, where EFs from burning conifers were higher than the model predicted. Removing the conifers' levoglucosan EFs results in $\log(EF_{levo})= -9.547$ (MCE)+8.041 and improves the correlation ($R^2=0.43$). Nonetheless, these measurements suggest compound EFs do depend to some extent on fuel type in addition to MCE. However, the spread

of measured EFs around the logarithmic fit in Figure 5 indicate a factor of 3 uncertainty in estimating EFs from MCE.

In addition to the well-known biomass burning particulate compounds shown in Figure 5, these measurements provide useful models to estimate EFs for hundreds of previously unreported compounds (not shown in Figure 5). These commonly detected compounds (i.e., found in >10 burns and occur in almost all fuel types) exhibited EFs that were inversely proportional to MCE. Regression parameters for compounds not displayed here are provided in the UCB-GLOBES FIREX

mass spectral library. Many of these compounds still remain unidentified or unknown (see Figure 3) but are now quantified as a function of fire conditions. Future work can be done to identify these compounds and ultimately, with the use of these regressions, estimate their contribution to I/SVOC mass in fresh smoke and model how they chemically transform in the atmosphere.

## 4 Conclusions

Smoke produced from burning a wide variety fuels, primarily from the western U.S. wildlands, was collected onto quartz fiber filters at the Fire Science Laboratory and analyzed for elemental and organic carbon. The organic carbon fraction was further separated, identified, classified, and quantified using TD-GC×GC-EI/VUV-HRToFMS with online derivatization.



Each separated compound's mass spectrum, *n*-alkane retention index, chemical family, and fuel source are reported here in a publicly available mass spectral library (UCB-GLOBES FIREX) for future comparisons and identification of biomass burning organic compounds in atmospheric samples. Between 10-65% of the I/SVOC mass for each burn could be specifically identified or placed into a chemical family. Fuels within the same type tended to exhibit similar mass fractions,

regardless of fire condition (as quantitated modified combustion efficiency, MCE). For example, similar unknown compounds accounted for ~90% of the total observed mass for the two woody debris burns (MCE=0.78-0.98). Conifers exhibited similar sugar and resin acid/diterpenoid mass fractions (out of total classified mass) of 30-50% and 2-20% respectively (MCE=0.90-0.95). Burns of coniferous duff (MCE=0.85-0.89) emitted higher classified mass fractions of methoxphenols (6-18%) than conifers. Peat, a semi-fossilized fuel, displayed a high classified mass fraction of non-cyclic

aliphatic/oxy compounds (52%). Shrubs showed the widest range in mass fractions, indicating fuels in this type were the most dissimilar.

Unlike mass fractions which depend primarily on fuel type, measured emission factors (EFs), classified into either organic carbon, chemical families, or specific compounds, primarily depended on fire conditions (MCE). Regardless of classification, EFs spanned 2-5 orders of magnitude from smoldering to flaming conditions. EFs were shown to follow an

inversely proportional relationship to MCE across the wide variety of all fuels studied. However, peat EFs for chemical families (except non-cyclic aliphatic compounds) and specific compounds were approximately a factor of 10 lower than fuels at similar MCE values. This is likely due to significant differences in fuel structure of peat. Furthermore, conifers exhibited higher sugar (factor of 5) and levoglucosan (factor of 3) emissions compared to other fuels within the same MCE range. This indicates that fuel type and specific fuels plays some role in the EFs, though more minor compared to MCE. This

is particularly true for nitrogen species and fuel-specific tracer compounds, i.e. compounds that are only emitted from a particular fuel, which will be discussed in a forthcoming paper. However, in general, EFs for these particulate compounds primarily depend on MCE and can be estimated from the fire conditions.

To provide modelers with useful relationships in estimating particle-phase I/SVOC emissions, logarithmic fits were applied to the measured EFs as a function of MCE. These regression models can be used to approximate EFs of I/SVOCs or their

chemical families from average MCE of real wildfires. For example, comparison with Liu et al. (2017) shows the estimated particulate organics from the regression model to be within a factor of 2-3 from those measured in that study. The comparison between predicted and previously measured EFs is affected by methodology, concentration regime, and the definitions of I/SVOC. Regardless, these regression models provide approximate EFs (within a factor of 3) of numerous chemical families and organic species as solely a function of fire conditions across a wide variety of fuels. These regressions

will allow modelers and other experimentalist to better define the chemical composition of smoke particles emitted from wildland burns in the western U.S. and potentially other parts of the world.





## 5 Data Sets

UCB-GLOBES can be downloaded at the Goldstein website, https://nature.berkeley.edu/ahg/data/MSLibrary/. The specific library for FIREX is FSL_FIREX2016_vX.msp  where X is the version number. The library contains information on all separated compounds observed during the FSL FIREX campaign in 2016 and will be periodically updated as compounds are matched across other campaigns.

## 6 Author Contributions

CNJ, LEH, VS, RJY, NMK, KCB, and AHG formulated the science question and designed the experimental setup. CNJ, LEH, VS, RJY, and AEF collected the data at FSL. CNJ and AHG analyzed the I/SVOC data. VS and RJY analyzed the CO and $CO_2$ data. RW organized the ALS campaign. CNJ wrote the manuscript with all authors contributing comments.

## 7 Acknowledgements

This work was supported by NOAA (NA16OAR4310107, NA16OAR4310103, and NA16OAR4310100) to UCB, UCR, and UM. CNJ acknowledges support from NSF PFS (AGS-1524211). Authors thank the staff at FSL and organizers of FIREX. The Advanced Light Source provided the VUV light and is supported by DOE. Special thanks to Bruce Rude and Dr. Kevin Wilson at LBNL for their assistance during the beamline campaign.

## 8 Competing Interests:

Authors declare that they have no conflict of interest.

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



**Table 1 List of fuels analyzed with the number of compounds separated and quantified from each burn. Note conifer fuel type refers to a realistic mixture of a coniferous ecosystem unless otherwise noted.**

| Fuel Description | Burn # | Fuel Type | Number of Compounds | MCE |
|---|---|---|---|---|
| Engelmann Spruce | 9 | Conifer | 714 | 0.9334 |
| Engelmann Duff | 12 | Coniferous Duff | 751 | 0.859 |
| Ponderosa Pine Rotten Log | 13 | Woody Debris | 709 | 0.9778 |
| Ponderosa Pine Litter | 16 | Coniferous Litter | 687 | 0.9607 |
| Engelmann Spruce Canopy | 17 | Conifer | 403 | 0.8953 |
| Douglas Fir Litter | 22 | Coniferous Litter | 585 | 0.9501 |
| Engelmann Spruce Duff | 26 | Coniferous Duff | 398 | 0.8474 |
| Manzanita Canopy | 28 | Shrub | 679 | 0.9789 |
| Douglas Fir Rotten Log | 31 | Woody Debris | 776 | 0.7785 |
| Manzanita Canopy | 33 | Shrub | 570 | 0.9788 |
| Engelmann Spruce Duff | 36 | Coniferous Duff | 596 | 0.8773 |
| Ponderosa Pine | 37 | Conifer | 811 | 0.9403 |
| Lodgepole Pine Canopy | 40 | Conifer | 444 | 0.9231 |
| Lodgepole Pine | 42 | Conifer | 634 | 0.9524 |
| Chamise Canopy | 46 | Shrub | 128 | 0.9566 |
| Subalpine Fir | 47 | Conifer | 596 | 0.9396 |
| Excelsior | 49 | Wood | 173 | 0.9712 |
| Yak Dung | 50 | Dung | 515 | 0.9016 |
| Peat, Kalimantan | 55 | Peat | 392 | 0.8405 |
| Subalpine Fir Duff | 56 | Coniferous Duff | 522 | 0.8874 |
| Rice Straw | 60 | Grass | 288 | 0.951 |
| Excelsior | 61 | Wood | 230 | 0.9508 |
| Bear Grass | 62 | Grass | 656 | 0.9036 |
| Lodgepole Pine | 63 | Conifer | 834 | 0.938 |
| Jeffery Pine Duff | 65 | Coniferous Duff | 472 | 0.8833 |
| Sage | 66 | Shrub | 328 | 0.9191 |
| Juniper Canopy | 68 | Conifer | 522 | 0.9293 |
| Kiln-Dried Lumber | 70 | Wood | 209 | 0.953 |
| Ceanothus Canopy | 74 | Shrub | 97 | 0.9748 |





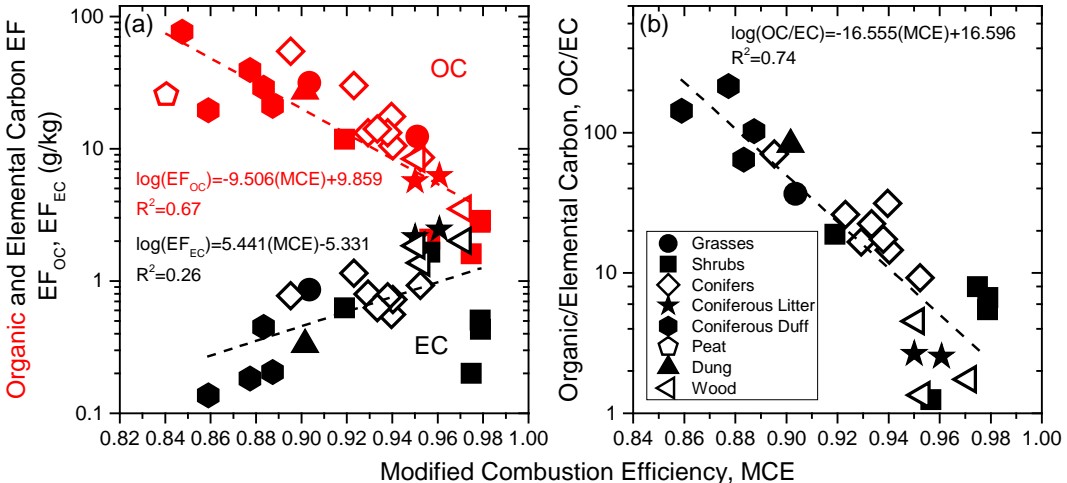

**Figure 1 (a) Measured organic and elemental carbon (OC and EC, respectively) as a function of modified combustion efficiency (MCE) and (b) OC/EC as a function of MCE. Symbols indicate the different fuel types.**



**Figure 2 Two-dimensional chromatogram of smoke collected from burning lodgepole pine (burn 63). First dimension separates compounds by their volatility and second dimension by their polarity. Each point (~800 in total) represents a separated compound with the colors signifying the compound's classification. Size of a point approximately scales with its emission factor.**





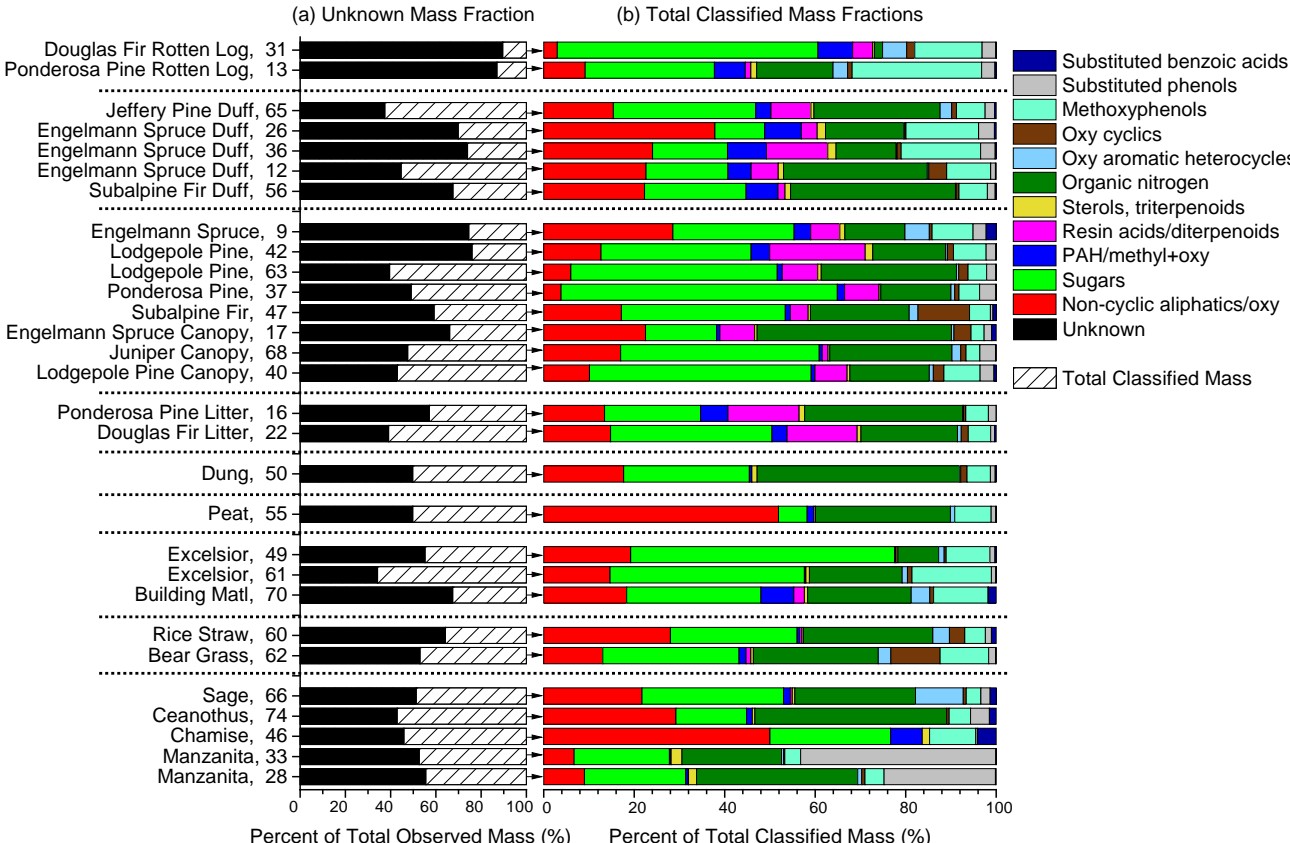

**Figure 3 (a) Contributions of unknown mass to the total observed mass for the 29 analyzed burns. (b) Mass fractions for each chemical family compared to total classified mass. Fuels are grouped by type and numbers after fuel name indicate the burn number during the FIREX campaign.**



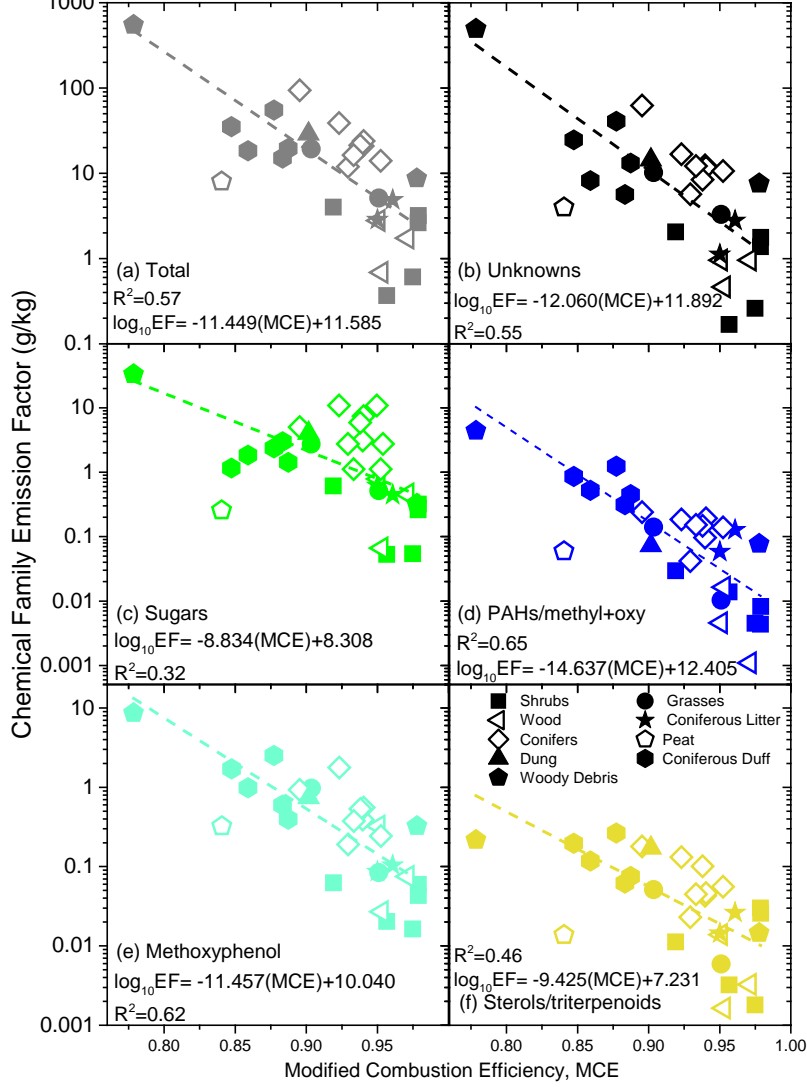

**Figure 4 Summed emission factors (EFs) within a chemical family for each burn as a function of modified combustion efficiency (MCE). Each panel depicts a different family with (a) total observed I/SVOC EF, (b) unknowns, (c), sugars, (d) Polycyclic aromatic hydrocarbons (PAHs, including methylated and oxygenated forms), (e) methoxyphenols, and (f) sterols/triterpenoids. Dashed lines represent a log fit of the form log(EF) inversely proportional to MCE. Symbols denote different fuel types.**





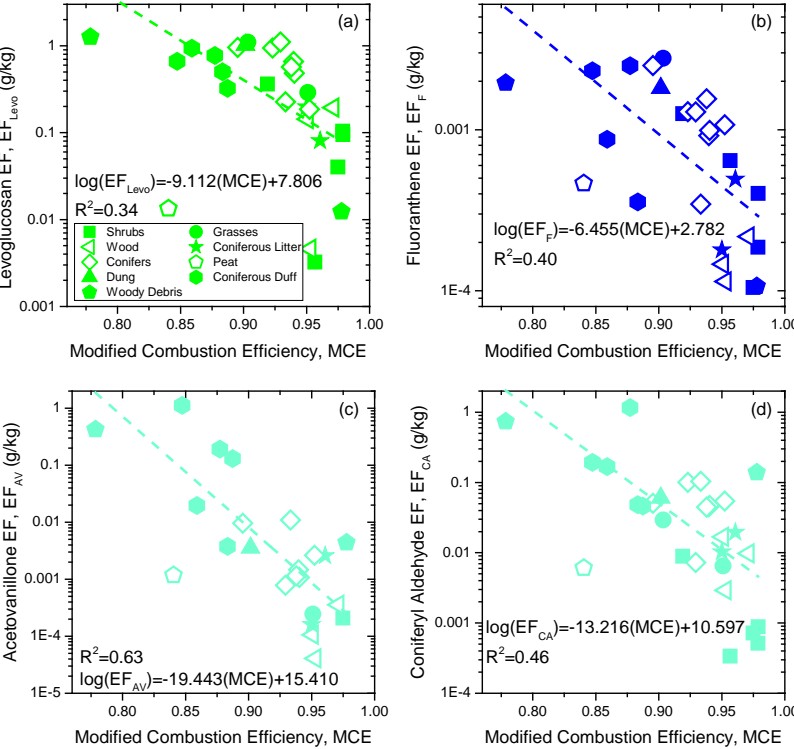

**Figure 5 Emission factors (EFs) of (a) levoglucosan (sugar), (b) fluoranthene (PAH), (c) acetovanillone (methoxyphenol), (d) coniferyl aldehyde (methoxyphenol) for various fuel burns as a function of MCE. Dashed lines indicate a log fit of the form log(EF) inversely proportional to MCE. Note, peat (open pentagon) is not included in the fit. Different symbols represent fuel categories.**