# Peer review of "Speciated and total emission factors of particulate organics from burning western U.S. wildland fuels and their dependence on combustion efficiency"

_Atmospheric Chemistry and Physics, 2018_

## Referee Comment (RC1) · Anonymous Referee #1 · 18 Oct 2018

This paper presents measurements of particulate organic compounds made during the FIREX lab campaign with a novel multi-stage measurement technique. The compounds are identified or assigned to functional groups where possible. The dependence of the emission factors of OC, EC, and the different compound classes and individual compounds with modified combustion efficiency and fuel type are analyzed.

This is an important piece of work that will further our efforts to understand the chemistry of organic aerosols from biomass burning smoke. The work appears to have been carefully done and important uncertainties and caveats are made clear. The conclu-

sions are generally justified by the results presented. The paper, tables, and figures are also generally clear and well presented.

I have no major concerns about the manuscript. Below I discuss a few minor issues that I would like the authors to address, and a few typos that need to be fixed.

Minor Comments:

P5, L31 and elsewhere: The numbers presented for the linear fits in this paragraph (including on page 6) do not match the numbers in Figure 1. Which set are correct? Please double-check all the numbers in the text to confirm they are consistent with the latest analysis of the data.

P7, L9-18 and Figure 2: This may be outside the scope of this paper, but would it be possible to map the two axes of Figure 2 to the saturation vapor concentration, O/C ratio, or the hygroscopicity "kappa" parameter? If so, that would help modelers use this data more directly.

P8, L1-3: You mention the uncertainty in the organic nitrogen compounds, but what about the uncertainty in the other EFs? How should those be treated?

P8, L10-14: The "shrub" class has the most variation in EFs, and I'm wondering if that's because the plants also have the most biological diversity in that class? I could see where all pines are basically the same but shrubs can be very different from one another.

P8, L17 and elsewhere: I agree that much of the variability in the EFs is due to MCE, but I think you understate the role of fuel type. It looks like the regression line is always low for conifers and always high for peat. I'd be curious what the effect of including fuel type as a factor variable in the linear regression would be (https://stats.idre.ucla.edu/r/modules/coding-for-categorical-variables-in-regression-models/) and if it would improve the fit.

P9, L15: I don't understand the statement that "the predicted total I/SVOC EFs are on

average higher than the measured EFs by a factor of 2." Isn't the point of a regression fit that "on average" the predicted value is the same as the measured value? How should I interpret this statement and the statements about the different fuel types?

P11, L17: I think this is the first time you discuss that the fuel structure of peat may be responsible for the difference, and I'm not sure you have any evidence for that statement, so I'd remove it from the conclusions.

Figure 2 caption: "Size of a point approximately scales with its emission factor" – is this a quantitative mapping from a function of some sort? It's be nice if the supplemental data explained how the size of the point relate to EF, or if you added a point size scale to the legend.

Table S6: That is a lot of significant figures given the error. Is there a reason you reported so many digits?

Typos:

P3, L10: Consider changing to "Therefore, a better representation"?

P5, L7: "laser transmittance laser"?

Section 3.2 heading and elsewhere: You don't need a colon at the end of headings

Figure 4f: The other panels listed the R2 and equation below the chemical family name, but it is listed at the bottom here. Please make consistent.

---

## Referee Comment (RC2) · Anonymous Referee #2 · 30 Oct 2018

Jen et al. have speciated particles and vapors from emissions of laboratory fires representative of those found in the Western US, as conducted at the Fire Sciences Lab in Missoula, MT. They performed 2D gas-chromatography/mass spectrometry to speciate a significant fraction of compounds from a whole range of organic families. Additionally, they were also able to develop log-linear regressions of the emission factors for these compounds with modified combustion efficiency (MCE) to aid development of fuel- and phase-specific emissions from fires in the Western US.

The study is well motivated, the methods are appropriate, and the manuscript is well

written. I had a few major comments surrounding the methods and data analysis. Regardless of my comments, I believe the speciation data from this study should help with modeling efforts to supplement the multi-agency ground, aircraft, and satellite-based studies involving fires in the United States (e.g., WE-CAN, FIREX-AQ). I would like to recommend publication of this study in Atmospheric Chemistry and Physics after the authors have responded to the following major and minor comments.

Major comments:

1. Identification, Page 6, Section 3.2: Of the 3000 compounds measured across the 29 fires, 149 seem to be positively identified. These probably have the highest certainty amongst the speciated compounds. What fraction of the total speciated and total mass do these represent? I am sure they probably change with fuel type but it would still be nice to know the range and some basic statistics (mean, standard deviation). For the remaining (3000-149) compounds, the authors refer to the SI for a more complete description of the methodology used to identify these compounds. It seems like Section 5 in the SI is what the authors are referring to. I found this description to be unsatisfactory and I am not sure this is a useful guide for readers if there were to replicate your methodology for their own work. What fraction of the total speciated and total mass do these remaining compounds account for, resolved by identified and unidentified? Finally, are the methods described herein common to analysis of GC/MS data and those of this research group? If they are, it would be beneficial to cite the group's earlier work in Section 3.2.

2. Calibration, Page 5, lines 13-18: Has the calibration technique described here been validated to work? If yes, can you cite the most recent literature? If not, would it be possible to split the dataset to validate this technique? How well would it work? Also, what are typical uncertainties in using a projected calibration (e.g., nearest sugar standard or nearest eluting standard) to calculate masses?

3. Fuel type as covariate, Sections 3.3-3.5: Despite the authors repeatedly saying

that fuel type was an important covariate, they failed to account for it in the regression models and limited their modeling to one with just a single covariate (i.e., MCE). The regression models in Figures 4 and 5 clearly show that the model when blanket-ly applied to any fuel can over/underestimate the emissions for certain types of fuels. Please consider using fuel type as a covariate to see if the regression model can be improved.

4. OC for mass closure, Page 7, lines 9-10: How did the final observed mass on the filters compare to the OC measurements? Wouldn't the OC be the gold standard to test for mass closure? If it is, shouldn't Figure 3 be done by normalizing with OC? Further, can the mass distribution from OC1 through OC4 be another useful constraint on the identification and calibration techniques since the 1 through 4 OC types are crude approximations for decreasing vapor pressure species?

5. Gas/particle partitioning: Were C* identified/developed for these species? What phase are these species expected to be in inside a fire plume or near background concentrations of organic aerosol? The C* for the species could be provided in the SI.

Minor comments:

1. Page 4, lines 10-11: If the quartz filters were the only ones analyzed in this study, I would state that after this sentence.

2. Page 4, line 18: Of the 75 fires, were only 29 sampled? Was there a reason the others were not?

3. Page 5, line 11: Why is a data inversion needed and what is it? Mention briefly in the main text.

4. Page 7, lines 3-5: Does this library also contain the emission factors for all the species measured in this study? This would be a useful resource to share with the community. It would also be beneficial to list the emission factors for species by fire that contribute a significant amount of the total observed mass in the SI (e.g., levoglucosan).

5. Page 8, lines 13-14: Can this final point about similarity within fuels be made statistically?

6. Page 9, line 14: Is 'accuracy' the right word here? Since you are testing the fit to the data, you are looking at the 'goodness-of-fit'.

7. Figures 4 and 5: Consider adding a factor of 2, 5, or 10 envelope on here to bound the deviation of the data from the fit.

8. SI: The Selimovic et al. citation seems to have shown up as both the ACPD and ACP paper. Please correct.

9. SI, page 12: A sentence in this section says 'see SI for more details'. Self-referencing?

---

## Author Comment (AC1) · 12 Dec 2018

We thank the reviewers for carefully reading and critically thinking about this paper. Below are our responses to the reviewers' comments. Text from the main paper or supporting information are provided in italics with changes in the manuscript's text highlighted in yellow. Reviewer comments are bold and our responses are normal typeface.

Reviewer 1:
**This paper presents measurements of particulate organic compounds made during the FIREX lab campaign with a novel multi-stage measurement technique. The compounds are identified or assigned to functional groups where possible. The dependence of the emission factors of OC, EC, and the different compound classes and individual compounds with modified combustion efficiency and fuel type are analyzed. This is an important piece of work that will further our efforts to understand the chemistry of organic aerosols from biomass burning smoke. The work appears to have been carefully done and important uncertainties and caveats are made clear. The conclusions are generally justified by the results presented. The paper, tables, and figures are also generally clear and well presented. I have no major concerns about the manuscript. Below I discuss a few minor issues that I would like the authors to address, and a few typos that need to be fixed.**

**P5, L31 and elsewhere: The numbers presented for the linear fits in this paragraph (including on page 6) do not match the numbers in Figure 1. Which set are correct? Please double-check all the numbers in the text to confirm they are consistent with the latest analysis of the data.**

We have corrected these typos and have gone through the rest of the text to ensure the written slopes match those of the figures.

*"EFs for OC and EC generally follow a logarithmic relationship such that log(EF$_{OC}$) is inversely proportional to MCE (slope of -9.506) and log(EF$_{EC}$) is directly proportional (slope 5.441). Comparison of the slopes suggests that decreasing MCE of a fire will produce an increasing amount of OC compared to EC. This is further confirmed by examining the ratio of OC to EC (OC/EC) with MCE. Figure 1(b) illustrates how OC/EC sharply increases with more smoldering fire conditions (slope of -16.555)."*

**P7, L9-18 and Figure 2: This may be outside the scope of this paper, but would it be possible to map the two axes of Figure 2 to the saturation vapor concentration, O/C ratio, or the hygroscopicity "kappa" parameter? If so, that would help modelers use this data more directly.**
Though we completely agree that displaying the chromatogram in terms of C* or kappa would be helpful for modelers, this would fall outside the scope of this study. Converting GC retention times to C* has done before for underivatized compounds (i.e. non-polar compounds); we refer

the reviewer to Isaacman et al. (2011). However, this conversion method becomes more difficult and more uncertain for derivatized, polar compounds as each added trimethylsilyl group will alter the measured retention time and thus C*. A more accurate approach to estimate C* for these compounds can be found in Hatch et al. (2018).

Obtaining O/C ratio from our measurements would require assigning parent masses for all the observed compounds. Previous work from the Goldstein group has done this for underivatized organic compounds, i.e., mostly non-polar compounds with minimal functionality (Isaacman et al., 2011). However, compounds emitted from biomass burning tend to be highly functionalized with groups such as hydroxyl, amino, and thiols. To detect these compounds using GCxGC entails derivatization. Therefore, converting parent mass to a molecular formula would require knowledge of the number of derivatized functional groups and the identity of the polar functional groups. This is knowledge we do not have for the vast majority of compounds. A much more detailed examination of the fragmentation pattern observed in the VUV (i.e., soft ionization) mass spectra could be done to help determine the number of derivatization groups and potential functionality. However, this would be a time-intensive process and would still have extremely high uncertainty. As a result, we believe extracting O/C ratios from our observations would be would be far outside the scope of this paper but may be worthwhile in the future.

**P8, L1-3: You mention the uncertainty in the organic nitrogen compounds, but what about the uncertainty in the other EFs? How should those be treated?**
We have clarified that the uncertainty for classified, unidentified, non-organic nitrogen compounds is ~30% and expanded the discussion on why this value is lower than previously reported from our group (see (Zhang et al., 2018)). In contrast, we do not have much chemical information for the unknown compounds in general. As a result, we estimate the uncertainty to be a factor of 2. These uncertainties are stated in the SI where the calibration method is discussed more in depth.

*Mass loading calibration curves were determined by measuring the instrument's response to varying amounts of 99 standard compounds typically found in biomass burning organic aerosol particles. We estimate the systematic uncertainty in the mass loadings for the unknown compounds at a factor of 2. Unidentified but classified compounds exhibited lower uncertainty due to similarities in instrument response to standards within the same family. To illustrate this reduction of uncertainty, we examine compounds with a RI of in the range of 1800-1900. Compounds that elute in this region include sugars, PAHs, aliphatics, and organic nitrogen. Their associated slopes from their mass loading calibration curves and compound family are provided in Table S2. Slopes within compound families are more similar than between families. For example, sugars exhibit slopes on average of 0.19 (not all shown in Table S2) whereas aliphatics have slopes of 1.1. An unclassified sample compound that elutes near myristic acid and galactose could be converted to mass loadings using either the slopes of myristic acid (0.43) or galactose (0.004). Depending which is chosen, the estimated mass*

*loading of this unclassified compounds could range over three orders of magnitude. However, if this sample compound were classified as a sugar, then the estimated mass loadings would be significantly higher and more in-line with the how typical sugars respond in the instrument. Our observations using various standard compounds indicate this calibration technique primarily lowers the uncertainty of more polar compounds to ±~30%.*

*Sampled compounds that exactly matched a standard compound have a lower uncertainty of ~±10% that is primarily due to instrument variation. Since the same data inversion factor was applied to the same observed compound across all samples, these systematic uncertainties do not affect the trends observed in this study but may affect the mass fractions each compound contributes to the total observed mass from a burn.*

**P8, L10-14: The "shrub" class has the most variation in EFs, and I'm wondering if that's because the plants also have the most biological diversity in that class? I could see where all pines are basically the same but shrubs can be very different from one another.**

Yes, we agree with the reviewer that the shrubs likely exhibit the widest diversity in plant chemical composition compared to the other fuel classes. Differences between fuel chemical composition, as also seen with peat, leads to a wider range in observed EFs. We have added this to the main text.

*Shrubs (MCE=0.92-0.98) exhibited the largest ranges in chemical family mass fractions (e.g., 0-42% organic nitrogen compounds and 2-43% substituted phenols), suggesting that plants in this fuel type are less similar to each other than coniferous fuels. This may be due to a wider range of plant chemical composition for shrubs than for the other fuel types. Overall, the I/SVOC mass fractions tend to be more similar for fuels within a fuel type with the most variation exhibited for fuel mixtures and shrubs.*

**P8, L17 and elsewhere: I agree that much of the variability in the EFs is due to MCE, but I think you understate the role of fuel type. It looks like the regression line is always low for conifers and always high for peat. I'd be curious what the effect of including fuel type as a factor variable in the linear regression would be (https://stats.idre.ucla.edu/r/modules/coding-for-categorical-variables-inregression-models/) and if it would improve the fit.**

We agree with both reviewer 1 and 2 about the partial dependence of EFs on fuel type. However, we believe including fuel type as a co-variant would not be very useful for several reasons. (1) The number of plants within each fuel type except for the conifers and coniferous duff is low. This is simply due to the specific fuel samples chosen for burning during the Fire Lab studies.

We would require more burns of a broader range of plants within each fuel type in order more definitely establish fuel type as a co-variant. (2) Fuel type and MCE are not independent variables. Note from Figure 3, fuels within the same fuel type fall within a characteristic MCE window. This indicates that fuel type (and likely other factors such as moisture and geometry) and MCE are not independent variables, which would also complicate regression modeling. Though this suggestion from reviewer 1 and 2 is logical, we believe to accomplish this would be outside the scope of this study and could be a separate study in the future.

**P9, L15: I don't understand the statement that "the predicted total I/SVOC EFs are on average higher than the measured EFs by a factor of 2." Isn't the point of a regression fit that "on average" the predicted value is the same as the measured value? How should I interpret this statement and the statements about the different fuel types?**
We agree this is a confusing statement. We have changed it to clarify our point which is to highlight the differences between the model and particular fuel types. In addition, we have decided to present as a ratio of model to observed so the numbers have changed a bit.

*The goodness-of-fit for the multi-fuel regression models can be evaluated by comparing the predicted EFs to those measured for the various fuel types in this study and others (Liu et al., 2017). As evident in Figure 4a, the predicted total I/SVOC to observed EFs are between 0.7-11 times higher for shrubs, 0.90-0.97 for grasses, 0.22-0.74 for conifers, 0.63-3.0 for coniferous duff, and 0.28-0.85 for woody debris.*

**P11, L17: I think this is the first time you discuss that the fuel structure of peat may be responsible for the difference, and I'm not sure you have any evidence for that statement, so I'd remove it from the conclusions.**

We actually mention this is section 3.4 and cite Stockwell et al. (2016) since we are not the first to observe differences of peat with other fuels.

**Figure 2 caption: "Size of a point approximately scales with its emission factor" – is this a quantitative mapping from a function of some sort? It's be nice if the supplemental data explained how the size of the point relate to EF, or if you added a point size scale to the legend.**

Though the points do scale with EFs, we had to make corrections to the floor and ceiling limits of point sizes. This was done to prevent some points from dominating the entire area of the chromatogram and the minute points from fading from view. As a result, we cannot add a useful point scale size to the legend. However, we do mention in the main text that all the EFs as a function of MCE are provided in UCB-GLOBES. We have added this information to the supporting information (in section 3).

*EFs for all observed compounds are provided in the open access FIREX data archive (see Data Sets of the main paper). Figure 2 illustrates the EFs for the observed compounds from a lodgepole pine burn. The marker sizes approximately scale with EFs. However, corrections were made to the floor and ceiling limits of the marker sizes. This was done to prevent some markers from dominating the entire area of the chromatogram and the minute points from fading from view.*

**Table S6: That is a lot of significant figures given the error. Is there a reason you reported so many digits?**

We have cut down the significant digits to better reflect the standard deviation.

|  | Shrubs | Grass | Wood | Coniferous Litter | Conifers | Peat | Dung | Coniferous Duff | Woody Debris |
|---|---|---|---|---|---|---|---|---|---|
| **Unknown** | 50%, 5% | 60%, 6% | 50%, 14% | 50%, 9% | 60%, 13% | 50% | 50% | 60%, 15% | 88%, 1% |
| **Non-cyclic aliphatics/oxy** | 10%, 9% | 8%, 2% | 8%, 2% | 7%, 2% | 6%, 2% | 26% | 9% | 9%, 2% | 1%, 0% |
| **Sugars** | 10%, 3% | 12%, 2% | 20%, 8% | 15%, 6% | 20%, 10% | 3% | 14% | 10%, 6% | 5%, 1% |
| **PAH/methyl+oxy** | 1%, 1% | 0%, 0% | 1%, 1% | 2%, 0% | 1%, 0% | 1% | 0%, | 2%, 0% | 1%, 0% |
| **Resin acids /diterpenoids** | 0%, 0% | 0%, 0% | 0%, 0% | 8%, 1% | 3%, 2% | 0% | 0% | 3%, 2% | 0%, 0% |
| **Sterols, triterpenoids** | 1%, 0% | 0%, 0% | 0%, 0% | 1%, 0% | 0%, 0% | 0% | 1% | 0%, 0% | 0%, 0% |
| **Organic nitrogen** | 13%, 8% | 12%, 1% | 8%, 4% | 14%, 1% | 10%, 5% | 15% | 22% | 11%, 6% | 1%, 1% |
| **Oxy aromatic heterocycles** | 1%, 2% | 1%, 0% | 1%, 0% | 0%, 0% | 1%, 0% | 0% | 0% | 0%, 1% | 0%, 0% |
| **Oxy cyclics** | 0%, 0% | 3%, 2% | 0%, 0% | 1%, 0% | 1%, 1% | 0% | 1% | 1%, 1% | 0%, 0% |
| **Methoxyphenols** | 3%, 1% | 3%, 2% | 7%, 3% | 3%, 0% | 2%, 1% | 4% | 3% | 4%, 1% | 3%, 1% |
| **Substituted phenols** | 7%, 0% | 1%, 0% | 0%, 0% | 1%, 0% | 1%, 1% | 1% | 1% | 1%, 0% | 0%, 0% |
| **Substituted benzoic acids** | 1%, 1% | 0%, 0% | 0%, 0% | 0%, 0% | 0%, 0% | 0% | 0% | 0%, 0% | 0%, 0% |
| **Average MCE.** | 0.958 | 0.898 | 0.958 | 0.955 | 0.931 | 0.840 | 0.902 | 0.871 | 0.878 |

**Typos:**
**P3, L10: Consider changing to "Therefore, a better representation"?**

We have done this.

*Therefore, a better or estimable representation of the chemical composition in smoke particles within models requires condensing the information from molecular-level speciation into useable relationships that correlate typical particle composition to a measurable burn variable.*

**P5, L7: "laser transmittance laser"?**
We removed the second laser.

**Section 3.2 heading and elsewhere: You don't need a colon at the end of headings**
We have removed them.

**Figure 4f: The other panels listed the R2 and equation below the chemical family name, but it is listed at the bottom here. Please make consistent..**

We have tried to keep the labels consistent but were unable for some. This is due to space constraints where the label would overlap with a data point. Therefore, we had to switch the order of the label around to make it legible.

Reviewer 2:

**Jen et al. have speciated particles and vapors from emissions of laboratory fires representative of those found in the Western US, as conducted at the Fire Sciences Lab in Missoula, MT. They performed 2D gas-chromatography/mass spectrometry to speciate a significant fraction of compounds from a whole range of organic families. Additionally, they were also able to develop log-linear regressions of the emission factors for these compounds with modified combustion efficiency (MCE) to aid development of fuel- and phase-specific emissions from fires in the Western US. The study is well motivated, the methods are appropriate, and the manuscript is well written. I had a few major comments surrounding the methods and data analysis. Regardless of my comments, I believe the speciation data from this study should help with modeling efforts to supplement the multi-agency ground, aircraft, and satellite based studies involving fires in the United States (e.g., WE-CAN, FIREX-AQ). I would like to recommend publication of this study in Atmospheric Chemistry and Physics after the authors have responded to the following major and minor comments.**

**Major comments:**
**1. Identification, Page 6, Section 3.2: Of the 3000 compounds measured across the 29 fires, 149 seem to be positively identified. These probably have the highest certainty amongst the speciated compounds. What fraction of the total speciated and total mass do these represent? I am sure they probably change with fuel type but it would still be nice to know the range and some basic statistics (mean, standard deviation).**

This is an excellent suggestion. We have looked into this. The mass fractions of positively identified compounds range between 4-37% between the various burns (mean of 20% with a std of 9%). There does not appear to be a correlation of mass fractions of positively identified with

MCE. This is expected because which compounds are identified does not depend on combustion efficiency. We have added this quantification to the main text in section 3.2.

*Identified compounds account for 4-37% of the total observed organic mass (mean of 20% with a standard deviation of 9%).*

**For the remaining (3000-149) compounds, the authors refer to the SI for a more complete description of the methodology used to identify these compounds. It seems like Section 5 in the SI is what the authors are referring to. I found this description to be unsatisfactory and I am not sure this is a useful guide for readers if there were to replicate your methodology for their own work.**

We agree that there should be some form of algorithm developed to automatically classify compounds into chemical families. However, the signals in derivatized electron ionization mass spectra, retention times, and vacuum ultra-violet ionization mass spectra vary considerably between compounds within the same family. Close examination of all the available information and expert judgement allow identification of patterns within chemical families; some examples of the useful patterns are given in the section 5 of the. To our knowledge, no formal algorithm exists that is capable of combing through large chemical data set to organize compounds by families. There is clearly a need to develop such an algorithm but this falls outside the scope of this study.

**What fraction of the total speciated and total mass do these remaining compounds account for, resolved by identified and unidentified?**

The identified compounds (149 of them) account for 4-37% of the total observed mass. Classified compounds (~400 compounds), including identified compounds, account for 10-65% of the total observed mass (see Figure 3(a)). We did not measure the total mass which would also include inorganic species, black carbon, and non-volatile organic compounds (e.g., extremely low volatility organic compounds). These types of compounds would not thermally desorb off the filters and thus not be measured.

**Finally, are the methods described herein common to analysis of GC/MS data and those of this research group? If they are, it would be beneficial to cite the group's earlier work in Section 3.2.**
In general, our technique for identifying compounds is standard for GC/MS analysis in that we first compare the electron ionization mass spectra and first dimension retention index against the National Institute of Standards and Technology (NIST) mass spectral database. However, NIST MS Library contains only a small fraction of possible compounds and an even smaller fraction of possible derivatized compounds. As a result, our group developed a soft ionization technique to

help preserve the parent ion of these compounds for improved identification. This paper, Isaacman et al. (2012), has been cited in our main text. Furthermore, Worton et al. (2017a) have shown that even "great" matches with NIST mass spectral database (match factors above 900) still have a 14% chance of incorrect identification for underivatized compounds. Consequently, positive identification requires running a standard compound on the instrument to confirm identity. The method used to ID compounds here is a combination of comparing to NIST MS Library, parent ion mass, and standards and is shown in Table S1 in the ID Method column.

We have added the reference Worton et al. (2017) to the main text.

*From those compounds, 149 compounds were identified using a combination of matching authentic standards (STD), RI, EI mass spectrum (via NIST mass spectral database, 2014 version), and VUV parent and fragment mass ions.* True positive identification requires analyzing a standard compound on the instrument; however comparing the NIST match to parent mass determined from VUV mass spectrum analysis can also provide a level of identification (Worton et al., 2017b). Identified compounds account for 4-37% of the total observed organic mass (mean of 20% with a standard deviation of 9%). A table of these identified compounds with their identifying methods (e.g., standard matching, previous literature, or NIST mass spectral database), RI, 5 most abundant mass ions from the EI mass spectra, and fuel source(s) are given in Table S1.

**2. Calibration, Page 5, lines 13-18: Has the calibration technique described here been validated to work? If yes, can you cite the most recent literature? If not, would it be possible to split the dataset to validate this technique? How well would it work? Also, what are typical uncertainties in using a projected calibration (e.g., nearest sugar standard or nearest eluting standard) to calculate masses?**

The reviewer is correct in identifying the uncertainties involved with this calibration technique. The issue with calibrating unknown compounds is that assumptions must be made about their behavior in the instrument. Previous studies from our group have calibrated unknown compounds using the closest eluting (in both dimensions of the chromatogram) standard compound. We refer the reviewer to the SI of Zhang et al. (2018) and have added this reference to the text. Zhang et al. estimated the uncertainty to be ~40% though this value is likely higher for more polar compounds.

In an effort to reduce uncertainties, we have refined this approach to better include chemical information. We assume compounds of the same chemical family will behave similarly in the instrument. Thus, we decided to calibrate classified compounds with the nearest eluting standard compound from the same chemical family. To illustrate this reduction of uncertainty, we examine compounds with a first-dimension linear retention index of ~1800. Compounds that

elute in this region include sugars, PAHs, aliphatics, and organic nitrogen. Their associated slopes from their mass loading calibration curves and compound family are provided in the table below. Slopes within compound families are more similar than between families. For example, sugars exhibit slopes on average of 0.19 whereas aliphatics have slopes of 1.1. A sample compound that elutes near myristic acid and galactose may have estimated mass loadings over three orders of magnitude depending on which standard compound is used. However, if this sample compound were classified as a sugar, then the estimated mass loadings will be significantly lower and more in-line with the how typical sugars respond in the instrument. Our observations using various standard compounds indicate this calibration technique primarily lowers the uncertainty of more polar compounds (i.e., compounds that require derivatization) from previously unknown percentage to ~30%. Illustrative data for selected standards is in below table.

| Compound Name | 1st dimension retention index | 2nd Dimension retention time (s) | Slopes from Mass Calibration | Compound Family |
|---|---|---|---|---|
| Octadecane (C18) | 1831 | 0.260 | 1.70 | Aliphatic |
| Mannose | 1831 | 0.310 | 0.19 | Sugar |
| Anthracene | 1836 | 0.680 | 1.82 | PAHs |
| Pinitol | 1856 | 0.330 | 0.37 | Sugar |
| 5-Nitrovanillin | 1866 | 1.350 | 0.67 | Organic nitrogen |
| Myristic Acid (C14 acid) | 1879 | 0.380 | 0.43 | Aliphatic |
| Galactose | 1885 | 0.320 | 0.004 | Sugar |

We have added this discussion into the supporting information, section 3.

*Mass loading calibration curves were determined by measuring the instrument's response to varying amounts of 99 standard compounds typically found in biomass burning organic aerosol particles. We estimate the systematic uncertainty in the mass loadings for the unknown compounds at a factor of 2. Unidentified but classified compounds exhibited lower uncertainty due to similarities in instrument response to standards within the same family. To illustrate this reduction of uncertainty, we examine compounds with a RI of in the range of 1800-1900. Compounds that elute in this region include sugars, PAHs, aliphatics, and organic nitrogen. Their associated slopes from their mass loading calibration curves and compound family are provided in Table S2. Slopes within compound families are more similar than between families. For example, sugars exhibit slopes on average of 0.19 (not all shown in Table S2) whereas aliphatics have slopes of 1.1. An unclassified sample compound that elutes near myristic acid and galactose could be converted to mass loadings using either the slopes of myristic acid (0.43) or galactose (0.004). Depending which is chosen, the estimated mass*

*loading of this unclassified compounds could range over three orders of magnitude. However, if this sample compound were classified as a sugar, then the estimated mass loadings would be significantly higher and more in-line with the how typical sugars respond in the instrument. Our observations using various standard compounds indicate this calibration technique primarily lowers the uncertainty of more polar compounds to ±~30%.*

*Table S2 Example mass loading calibrations slopes for compounds in the RI=1800 range.*

| Compound Name | 1D RI | 2D retention time (s) | Mass Calibration Slopes | Compound Family |
|---|---|---|---|---|
| Octadecane (C18) | 1831 | 0.260 | 1.70 | Aliphatic |
| Mannose | 1831 | 0.310 | 0.19 | Sugar |
| Anthracene | 1836 | 0.680 | 1.82 | PAHs |
| Pinitol | 1856 | 0.330 | 0.37 | Sugar |
| 5-Nitrovanillin | 1866 | 1.350 | 0.67 | Organic nitrogen |
| Myristic Acid (C14 acid) | 1879 | 0.380 | 0.43 | Aliphatic |
| Galactose | 1885 | 0.320 | 0.004 | Sugar |

**3. Fuel type as covariate, Sections 3.3-3.5: Despite the authors repeatedly saying that fuel type was an important covariate, they failed to account for it in the regression models and limited their modeling to one with just a single covariate (i.e., MCE). The regression models in Figures 4 and 5 clearly show that the model when blanket-ly applied to any fuel can over/underestimate the emissions for certain types of fuels. Please consider using fuel type as a covariate to see if the regression model can be improved.**

The reviewer does point out an important observation of this study in that emission factors (EFs) of I/SVOCs does partially depend on fuel type. A notable example is coniferous fuels emitting high amounts of sugar compounds. Though the reviewer does suggest a valid way of improving the accuracy of our model by including dependence on fuel type, we have opted to not change our model for several reasons. First, relating EFs to MCE and fuel type is more complex than including fuel type as a covariant. Fuel type and MCE are not independent variables as fuel types can often burn within a characteristic range of MCE. For example, shrubs burned efficiently with MCE~0.97 and coniferous duff burned inefficiently with MCE~0.85. Furthermore, to include a robust analysis of EFs on fuel type would require significantly more emission samples within each fuel type that span a wider range of MCE. Though we conducted stack burns at the FIREX FSL campaign, we focused primarily on fuels found in the western US and only a handful of plants in each of the fuel types. We would need to burn similar fuels across a wider range of MCE and more fuels in general in order to better tease out the relationship of EFs with fuel type.

In real world modeling, the actual fuel type-specific measurements we present can be used if the fuels are known for a fire. However, the fuels, or mix thereof, are often unknown in which case our regression model provides a reasonable estimate for EFs of specific compounds or chemical families.

We have made this final point clearer in the conclusion of the main paper.

*To provide modelers with useful relationships in estimating particle-phase I/SVOC emissions, logarithmic fits were applied to the measured EFs as a function of MCE. These regression models can be used to approximate EFs of I/SVOCs or their chemical families from average MCE of real wildfires where fuel loadings, fuel types, and fuel mixtures are often unknown.*

**4. OC for mass closure, Page 7, lines 9-10: How did the final observed mass on the filters compare to the OC measurements? Wouldn't the OC be the gold standard to test for mass closure? If it is, shouldn't Figure 3 be done by normalizing with OC? Further, can the mass distribution from OC1 through OC4 be another useful constraint on the identification and calibration techniques since the 1 through 4 OC types are crude approximations for decreasing vapor pressure species?**

Organic carbon mass closure is the ultimate goal of speciated measurements of organic compounds in aerosol particles. We have carefully considered this during our data analysis however we stopped short of including an organic carbon to total observed I/SVOC comparison for two reasons: (1) converting organic carbon to organic aerosol is not trivial and (2) parent mass identification for all observed I/SVOCs is required. Organic carbon (OC) is often converted to organic mass (OM) using an empirically derived number between 1.4-1.7. Russell has shown that this number varies widely between samples collected from various locations (Russell, 2003). Aiken et al. provide ratios between 1.5-1.7 from Fire Lab burns of Lodgepole Pine and grass burns. The range of ratio values leads us to believe that we should determine our own OM/OC value as our TD-GCxGC VUV-EI/HRTOFMS can provide parent masses for all observed compounds. However, assigning parent masses using the VUV spectrum requires knowing the number of derivatized groups on the unknown organic compounds and which functional groups were derivatized (hydroxyl, amino, thiol). We currently have no method for determining these constraints. Furthermore, our technique sees a specific window of organic compounds (I/SVOCs) and does not include low volatility compounds which may account for a significant fraction of the organic carbon (~20% of total per May et al. (2013)). As stated in the main paper, collection onto quartz fiber filters likely includes gas-phase artifacts (i.e., VOCs) that would be seen by the OCEC analyzer but not by our GCxGC. We concluded from these reasons that we cannot easily compare OC to our total observed I/SVOC mass.

Comparing the OCEC thermograms to the retention time distributions of the GCxGC also cannot be done easily. This is due to the presence of VOCs on the filters which would impact the OCEC thermograms and the fact that we derivatize our compounds prior to GCxGC analysis. Derivatizing a compound will alter its volatility and therefore its retention time in the GCxGC.

With regards to OC being the gold standard: Studies have shown the measured OC and EC amounts (and thus the OC:EC ratio) is impacted by the measurement technique such as the transmittance and reflectance charring correction (see Chen et al. (2011) as an example). We hesitate to claim the OC measurement is a gold standard though it very likely has lower uncertainty values than our measured mass loadings from the GCxGC.

**5. Gas/particle partitioning: Were C\* identified/developed for these species? What phase are these species expected to be in inside a fire plume or near background concentrations of organic aerosol? The C\* for the species could be provided in the SI.**

Reviewer 1 mentions this same point. We completely agree that C\* is a very useful parameter for the community. However, converting retention time/index into C\* with derivatized compounds has not been done before and would likely lead to large errors in estimated C\*. We refer the reviewer to Hatch et al. (2018) for a more accurate method in determining C\*.

**Minor comments:**
**1. Page 4, lines 10-11: If the quartz filters were the only ones analyzed in this study, I would state that after this sentence.**
We have added this.

**2. Page 4, line 18: Of the 75 fires, were only 29 sampled? Was there a reason the others were not?**
Yes, 29 out of the 75 burns were analyzed. These 29 represent all the unique fuels burned, including several replicates and fuels from different collection locations to explore potential variability between burns of the same fuel. After examining the chromatograms of several replicate burns, we determined the variability between replicates was minor (except for fuels from a different location). Thus, we opted to not analyze the remaining 46 burns with the same level of detail as required for the presented analyses due to the large amount of time required. We have added this explanation to the text.

*Chromatograms from replicate burns showed minor variation thus the remaining 46 burns were not analyzed in detail.*

**3. Page 5, line 11: Why is a data inversion needed and what is it? Mention briefly in the main text.**

The GCxGC does not measure mass directly but rather provides a signal that is affected by matrix effects, instrument sensitivity, and other factors. Data inversion is needed to convert instrument signal to mass loadings (and subsequent emission factors). We have opted to change the text to remove confusion of using the word inversion.

*Full details of the data ==conversion to mass loadings and emission factors with== its associated uncertainties are provided in the SI with important steps outlined here.*

**4. Page 7, lines 3-5: Does this library also contain the emission factors for all the species measured in this study? This would be a useful resource to share with the community. It would also be beneficial to list the emission factors for species by fire that contribute a significant amount of the total observed mass in the SI (e.g., levoglucosan).**

Since numerous compounds are found in multiple burns, the library contains the EFs vs MCE relationships. We apologize for not including this in the library description and have now added it. These EF relationships have always been in UCB-GLOBES FIREX and now we have added it to the description.

*This spectral library is compatible with NIST MS Search and contains mass spectra, n-alkane RI, potential compound identification or chemical families, ==EFs as a function of fire conditions,== and fuel sources of all unique compounds detected from the 29 analyzed burns.*

*Each separated compound's mass spectrum, n-alkane retention index, chemical family, ==EF vs. MCE relationship,== and fuel source are reported here in a publicly available mass spectral library (UCB-GLOBES FIREX) for future comparisons and identification of biomass burning organic compounds in atmospheric samples.*

We have opted to not list the EFs for species that contribute a significant amount to the total observed mass in the SI because for most of the burns, these compounds are unknown and unclassified. However, we do believe this information is useful, so we have directed the reader in the "Data Sets" section to the open access data archives for NOAA FIREX where all of our observed EFs are given for each of the burns. This can be found at http://esrl.noaa.gov/csd/groups/csd7/measurements/2016firex/FireLab/DataDownload/.

*UCB-GLOBES can be downloaded at the Goldstein website, https://nature.berkeley.edu/ahg/data/MSLibrary/. The specific library for FIREX is FSL_FIREX2016_vX.msp  where X is the version number. The library contains information on all separated compounds observed during the FSL FIREX campaign in 2016 and will be periodically updated as compounds are matched across other campaigns. ==Observed emission factors for all of the observed compounds for each of the analyzed burns can be accessed for free==*

*through the NOAA FIREX data archives
([http://esrl.noaa.gov/csd/groups/csd7/measurements/2016firex/FireLab/DataDownload/](http://esrl.noaa.gov/csd/groups/csd7/measurements/2016firex/FireLab/DataDownload/))*.

**5. Page 8, lines 13-14: Can this final point about similarity within fuels be made statistically?**

This is a good suggestion though we believe this falls outside the scope of this study as we are more qualitatively comparing mass fractions between fuel types here. Our collaborators Lindsay Hatch and Kelley Barsanti at UC Riverside have two papers that more quantitatively compared compounds between fuel types: (Hatch et al., 2018) and a paper under review in EST.

**6. Page 9, line 14: Is 'accuracy' the right word here? Since you are testing the fit to the data, you are looking at the 'goodness-of-fit'.**
We agree and have changed this to goodness of fit.

*The goodness-of-fit for the multi-fuel regression models can be evaluated by comparing the predicted EFs to those measured for the various fuel types in this study and others (Liu et al., 2017).*

**7. Figures 4 and 5: Consider adding a factor of 2, 5, or 10 envelope on here to bound the deviation of the data from the fit.**

We have added a plus/minus factor of 2 bounds to figure 4 and 5.

[Figure]

**Figure 4** Summed emission factors (EFs) within a chemical family for each burn as a function of modified combustion efficiency (MCE). Each panel depicts a different family with (a) total observed I/SVOC EF, (b) unknowns, (c), sugars, (d) Polycyclic aromatic hydrocarbons (PAHs, including methylated and oxygenated forms), (e) methoxyphenols, and (f) sterols/triterpenoids. Dashed lines represent a log fit of the form log(EF) inversely proportional to MCE. The dotted lines represented a factor of 2 above and below the model. Symbols denote different fuel types.

[Figure]

**Figure 5** Emission factors (EFs) of (a) levoglucosan (sugar), (b) fluoranthene (PAH), (c) acetovanillone (methoxyphenol), (d) coniferyl aldehyde (methoxyphenol) for various fuel burns as a function of MCE. Dashed lines indicate a log fit of the form log(EF) inversely proportional to MCE. Dotted lines show a factor of 2 above and below the model. Note, peat (open pentagon) is not included in any of the fits. Different symbols represent fuel categories.

**8. SI: The Selimovic et al. citation seems to have shown up as both the ACPD and ACP paper. Please correct.**

We have fixed this.

**9. SI, page 12: A sentence in this section says 'see SI for more details'. Self-referencing?**

We have fixed this to refer to the next section.

**References Cited:**

Cheng, Y., He, K., Duan, F., Zheng, M., Du, Z., Ma, Y. and Tan, J.: Ambient organic carbon to elemental carbon ratios: Influences of the measurement methods and implications, Atmos. Environ., 45(12), 2060–2066, doi:10.1016/j.atmosenv.2011.01.064, 2011.

[revised manuscript text omitted]